# Association between Selenium Intake and Optimal Sleep Duration: A National Longitudinal Study

**DOI:** 10.3390/nu15020397

**Published:** 2023-01-12

**Authors:** Lingxi Zhao, Shengping Li, Yuzhao Zhu, Tiankun Wang, Yu Su, Zumin Shi, Yangchang Zhang, Yong Zhao

**Affiliations:** 1School of Public Health, Chongqing Medical University, Chongqing 400016, China; 2Research Center for Medicine and Social Development, Chongqing Medical University, Chongqing 400016, China; 3Research Center for Public Health Security, Chongqing Medical University, Chongqing 400016, China; 4Human Nutrition Department, College of Health Sciences, QU Health, Qatar University, Doha 2713, Qatar; 5Department of Epidemiology and Health Statistics, School of Public Health, Capital Medical University, Beijing 100069, China; 6Chongqing Key Laboratory of Child Nutrition and Health, Children’s Hospital of Chongqing Medical University, Chongqing 400014, China

**Keywords:** selenium intake, sleep, Chinese, adults, nutrition survey

## Abstract

Inconsistent findings have been discovered in studies examining the link between dietary selenium (Se) and sleep. Data were obtained from 17,176 people aged 20 and over who participated in the China Health and Nutrition Survey (CHNS) from 2004 to 2011. Face-to-face interviews were used to measure sleep duration in 2004, 2006, 2009, and 2011. To track dietary Se consumption, a 3-day, 24-h recall was undertaken. In the analysis, multilevel mixed-effects logistic regression was employed. The odds ratios (95% confidence intervals) of optimal sleep duration (7–9 h/day) in the regression of Model 4 were 1.00, 1.01 (0.89–1.15) and 1.19 (1.02–1.38) for the three tertiles of selenium consumption, respectively. Only overweight patients displayed a substantial positive connection between Se intake and the optimal sleep duration in the subgroup analysis. In summary, Se intake was significantly associated with optimal sleep duration.

## 1. Introduction

For humans, sleeping is as important as eating [1]. A healthy adult should have 7–9 h of sleep time per night [2]. Health issues may result from long-term insufficiency or excess sleep. Previous studies have discovered that persistent sleep deprivation is connected to physiological, cognitive, and psychomotor disorders, such as obstructive sleep apnea (OSA), the declines of memory, concentration, learning, and even depression [3,4]. Compared with 2010, the prevalence of sleep deprivation (sleep time < 7 h/day) of working American adults increased by 4.7% in 2018 [5]; however, little information is available regarding the effects of prolonged sleep and higher mortality. Although few studies of this kind exist, several studies have suggested that excessive sleep may be linked to detrimental health consequences, including obesity, stroke, and coronary heart disease [6,7,8]. Minerals, such as magnesium, iron, and selenium, have been discovered to have a substantial relationship with habitual sleep length [9]. 

Selenium (Se), a naturally occurring chemical element that has toxicological and nutritional effects, is attracting a lot of interest in biology and medicine [9,10]. Selenoprotein, an antioxidant active material, is crucial for reducing reactive oxygen species and pro-inflammatory biomarkers [11]. However, a projection estimated that approximately one out of seven people will have a low level of Se intake in the future [12]. Furthermore, according to a review, 39% to 61% of Chinese residents have a low daily intake of selenium (26–34 μg/day), which is below the WHO/FAO recommendation [13].

The antioxidant role of Se is important to improve sleep quality and sleep duration [14,15]. Recently, some studies have demonstrated a correlation between Se and sleep. A study assessed the relationship between sleep and dietary/nutritional variables through the National Health and Nutrition Examination Survey (NHANES, 2007–2008). It has shown that low Se intake was associated with short sleep (5–6 h) compared with normal sleep (7–8 h) [16]. Furthermore, a case-control study published in 2020 has found that Se may be an important defense mechanism amongst patients with OSA without any comorbidities [17]. Rachel et al. [18] also proposed that Se can be used to treat depression, which is the main symptom of sleep disturbance. Nevertheless, the debate about Se and sleep is ongoing. A multicenter prospective cohort study published in 2021 suggested that no significant association exists between toenail Se concentrations and sleep quality [19]. Moreover, an animal study has shown that Se compounds can act as an inhibitor of prostaglandin D_2_ (PGD_2_) to increase the incidence of arousal behavior [20]. These inconsistent findings could be related to Se status and different soil Se levels for the study populations [21,22].

Currently, no relevant study has been conducted on the Chinese population. Using the latest data from the CHNS, we aimed to evaluate the association between Se intake and sleep duration in the general Chinese population.

## 2. Methods

### 2.1. Study Design and Study Sample

Data were collected from the CHNS, which is a follow-up project conducted by the Carolina Population Center and the National Institute for Nutrition and Health (China). CHNS aimed to investigate the population’s state of health and nutrition. A multistage stratified cluster random sampling design was used to sample over 30,000 people from 7200 families across 15 provinces.

The CHNS was started in 1989; ten waves of surveys have been carried out. The survey on sleep started in 2004, so we used data from 2004 to 2011 to examine the relationship between Se intake and sleep duration based on inclusion and exclusion.

A total of 22,090 individuals who were 20 years old or more were selected in the first round of selection. Subsequently, participants who failed to report their dietary intake (*n* = 3906), abnormal calorie consumption each day (men: above 6000 kcal or under 800 kcal; women: over 4000 or under 600 kcal) (*n* = 96), pregnant or breastfeeding women (*n* = 91), individuals with implausible body mass index (BMI) < 14 or > 45 kg/m^2^, *n* = 7) (*n* = 599), missing data on income (*n* = 212), and participants with outliers of physical activity (*n* = 75) were excluded. Ultimately, 17,176 participants were enrolled, and 10,865 participants had optimal sleep duration at baseline (sleep time 7–9 h/day) (Figure 1). Both the National Institute of Nutrition and Food Safety and the University of North Carolina (USA) approved the survey (China). Each participant provided their explicit written consent. On the official CHNS website, the information for this study is freely accessible. (https://www.cpc.unc.edu/projects/china. Accessed on 18 April 2022).

### 2.2. Outcome Variable: Optimal Sleep Duration

Self-reported sleep duration was determined by the question, ‘How many hours of sleep do you usually get each day at daytime and nighttime (in hours)?’. Daily sleep duration was categorized into short (≤6 h), optimal (7–9 h), and long (≥10 h) based on the data distribution and sleep reality amongst the Chinese population. Optimal sleep duration was defined as >6 h and <10 h per day [2,23,24].

### 2.3. Exposure Variable: Se Intake

In each wave, a professional investigator documented every person’s food intake over three consecutive days. Researchers were required to maintain a daily food record, weighed the household’s supply of food and condiments, and accounted for food waste. Nutrient intakes, including Se and protein, were computed based on the average of 3-day food consumption data utilizing the Chinese Food Composition Table (CFCT). The dietary data module, which was employed by CFCT to decipher the food code, contained information on each respondent’s mean daily intake of selenium (in micrograms). To assess Chinese people’s food intake and nutrition, three different CFCT versions were applied. With regard to specific waves, the 1981 CFCT approved specifically for the 1989, 1991, and 1993 waves; the 1991 CFCT for the 1997 and 2000 waves; and the 2002/2004 CFCT for 2004, 2006, 2009, and 2011. Regarding the nutritious content of specific food items, the CFCT considered the region. Consequently, geographical characteristics were connected to selenium content in this study.

In order to determine the independent variable, we computed the cumulative Se intake. This approach effectively decreased individual variance and represented permanent dietary habits [15]. To give a simple example, a person’s cumulative means for 2005, 2006, and 2009 are (q + x)/2, (q + x + y)/3, and (q + x + y + z)/4, respectively, if his selenium consumption in 2004, 2005, 2006, and 2009 was q, x, y, and z. The most recent intake for 2004, 2005, 2006, and 2009 was q, x, y, and z, respectively. The baseline intake was q [25]. In addition, depending on all observations, we divided the intake into tertiles.

### 2.4. Covariates

Sociodemographic characteristics and health-related behaviors in the survey results were taken as covariables. Sociodemographic information included gender (male or female), age, education level (primary school and below, junior high school, and senior high school and above), and income (low, medium, and high). The examined health-related variables included smoking (yes or no), drinking alcohol (yes or no), activity level (light, moderate, or heavy), and BMI based on the self-reported height and weight. Overweight was defined as having a BMI ≥ 24 kg/m^2^, which was used to measure adiposity levels [26]. A systolic blood pressure of more than 140 mmHg, diastolic blood pressure more than 90 mmHg, or with known hypertension is considered hypertension. Physical activity level (MET) was determined via self-reported activities (including occupational, domestic, transportation, and leisure-time physical activity) and duration using the Adult Compendium of Physical Activities. Self-reported diabetes was marked as “yes” or “no”." Geographically, Heilongjiang, Liaoning, Shandong, and Henan were considered to be in the north, and Jiangsu, Hubei, Hunan, Guizhou, and Guangxi were considered to be in the south [27].

### 2.5. Statistical Analyses

Continuous variables were reported as mean ± standard deviation (SD) and were examined using ANOVA or Kruskal–Wallis test. Chi-square test results for categorical variables were provided as frequencies and proportions (%). Multilevel mixed-effect logistic regression analysis was used to investigate the relationship between cumulative mean Se consumption and sleep duration. The ideal sleep length was treated as a function of fixed effects of Se consumption, whereas individual participant variables were handled as random effects. In order to take into consideration interindividual variations in baseline measurements and the rate of change, random-effects components were fitted to the intercept and slope [28]. Multilevel model description for the survey on optimal sleep duration: $$\log(\frac{\Pi_{ij}}{1-\Pi_{ij}})=\gamma_{00}+\gamma_{10}time_{ij}+(\mu_{0j}+\mu_{1j}time_{ij}+e_{ij})$$

Level 1 (Within the individual) model: $$\log(\frac{\Pi_{ij}}{1-\Pi_{ij}})=\beta_{0j}+\beta_{1j}time_{ij}+e_{ij}$$

Level 2 (Between individuals) model:$$\beta_{0j}=\gamma_{00}+\mu_{0j},\;\beta_{1j}=\gamma_{10}+\mu_{1j}$$

*j* represents the identification number of the participant, *i* represents the number of measurements, γ_00_ represents the incidence of optimal sleep duration at baseline, *e_ij_* represents the difference between each measurement of individual, β_0*j*_ and β_1*j*_ denote the incidence of optimal sleep duration for subject *j* at baseline and the change in the incidence of optimal sleep duration for subject *j* over time, respectively.

Se intake in the first tertile was taken as the reference group. A set of multivariable models was built: Model 1, which adjusted for age, gender, fat, and energy intake; Model 2, which further adjusted for smoking, alcohol drinking, income, urban, education, and physical activity; Model 3, which further adjusted for BMI and hypertension based on Model 2; and Model 4 which further added for self-reported diabetes, energy, and fat intake. In subgroup analyses, cumulative mean Se intake was used in the mixed-effect logistic regression models, and the interaction of cumulative mean Se intake in the diet and covariates, including urbanization level, gender, smoking, region, and alcohol intake, was assessed by adding a cross-product term to the overall multilevel mixed effects logistic regression model.

## 3. Results

Eligibility was met by 17,176 individuals who were included in this study. Table 1 demonstrated that across the Se consumption tertiles, the mean ± SD of Se intake was 23.9 ± 6.1, 39.2 ± 3.8, and 68.0 ± 44.8, respectively. Energy, fat, protein, and carbohydrate intake increased across the tertiles of Se intake. Alcohol consumption and BMI were higher in participants who consumed high Se. Se intake was strongly correlated with urbanization, literacy, and household income, while the consumption of selenium decreased with age.

### 3.1. Association between Se Intake and Self-Reported Optimal Sleep Duration

Self-reported optimum sleep duration and Se intake were positively correlated (Figure 2). In the model, age, gender, and calorie consumption were taken into account (model 1), the odds ratios (ORs) (95% confidence interval (CI)) were 1.00, 1.12 (1.03–1.21), and 1.37 (1.25–1.49) across the food selenium consumption tertiles. Nonetheless, after adding additional factors (e.g., smoking, alcohol consumption, income, urban residence, education, physical activity level, BMI, hypertension, fat intake, protein intake, carbohydrate intake, and self-reported diabetes), this correlation weakened and the value of *p* for trend was 0.032 (model 4). In the fully adjusted model, the ORs (95% CI) across the tertiles of Se intake were 1.00, 1.01 (0.89–1.15), and 1.19 (1.02–1.38), respectively.

### 3.2. Subgroup Analyses of Selenium Consumption with the Optimal Sleep in China Nutrition and Health Study

Selenium intake was not interactively related to income, hypertension, gender, education, and urbanization in relation to optimal sleep duration. Nonetheless, the optimal sleep duration was significantly influenced by the interaction between Se daily consumption and BMI (*p* = 0.018) (Table 2). When analyzing the subgroups, a high correlation between Se intake and the optimal sleep duration was found only amongst overweight participants compared with participants with a BMI < 24.

## 4. Discussion

The average daily intake of selenium examined in this study ranged from 42.3 micrograms per day in 2004 to 43.3 micrograms per day in 2011. However, the majority of participants consumed less selenium than the China EAR (50 micrograms per day) [29]. The cumulative average Se intake was less likely to result in improper sleep duration and was favorably associated with the ideal sleep duration. Moreover, the consumption of selenium had remarkable interactions with BMI. Only people with a high BMI had a favorable correlation between Se intake and the ideal amount of sleep.

Our research showed that people who consumed large amounts of Se experience the best sleep. More and more people know that both very long and excessively short sleep durations are thought to be detrimental. Additionally, several researchers have discovered a connection between selenium and the length of sleep. For example, only a brief sleep period (5–6 h) was revealed by the NHANES to be negatively correlated with selenium consumption [30]. The same result has been found in a case-control study [30]. However, data from experiments performed in a laboratory environment differed from our findings. Some animal studies have demonstrated that decreased sleeping time is due to the inhibition of PGD synthase activity in the brain by SeCl_4_ [31]. The main source of selenium for both people and animals is food, and the main organic selenium species in cereal and vegetables are selenomethionine and selenocysteine, which differ from selenium chloride (SeCl_4_) [32].

Further studies are needed to discover how selenium intake affects human sleep. Sleep deprivation and poor sleep quality are linked to oxidative stress and antioxidant imbalance, which could increase the prevalence of chronic diseases [33,34]. Selenoenzymes are important antioxidants, and they may be particularly beneficial in the defense against oxidative stress [35,36]. Antioxidant consumption may influence sleep because sleep duration can be affected by pro-inflammatory cytokines [37,38]. A review has indicated that some pro-inflammatory cytokines are associated with increased body temperature, reduced nREM sleep duration, and increased wakefulness, thereby proving that having insomnia is connected to increased plasma cytokine levels [39]. The body produces many leukocytes and mast cells in the injured area as a result of the inflammatory response, which increases the generation and release of ROS by increasing the intake of oxygen [40]. Meanwhile, antioxidants show the strongest anti-inflammatory activities through the inhibition of inducible pro-inflammatory cytokines expression and ROS generation [41,42]. In summary, the intake of Se could increase the activity of antioxidant enzymes and inhibit ROS generation, providing reliable relief from insomnia and promoting healthy sleep.

The interaction between BMI and Se intake in terms of the ideal amount of sleep was interesting. In our study, the positive association between Se intake and optimal sleep duration was observed only in overweight people. The Wisconsin Sleep Cohort Study has demonstrated a U-shaped curvilinear association between sleep duration and BMI [43]. Dashti et al. [44] indicated robust causal effects of insomnia on higher BMI and, conversely, of higher BMI on snoring and daytime sleepiness. Additionally, bidirectional effects between sleep duration and daytime napping with obesity may exist. The third National Health and Nutrition Examination Survey in the US has further found that BMI was inversely associated with serum Se concentrations in men and women [44], inconsistent with our finding. The reason may be that we measured only food Se intake instead of Se biomarkers. Meanwhile, high BMI is associated with an inflammatory state, which reduces circulating Se [45,46]. In summary, having a BMI < 24 may be a standalone protective factor for getting the recommended amount of sleep. The interaction mechanisms between overweight and Se to optimal sleep duration require further exploration.

Our study has a number of advantages that should be emphasized. First, we used four cycles of CHNS data. The study sample represented the general population. The sample size was large and improved the precision of estimates. Second, the long-term relationship between dietary selenium intake and sleep has not been extensively studied in the literature. Third, in order to provide a reliable estimate of long-term Se consumption, this study used a prospective design with numerous rounds of nutritional evaluations on the basis of three-day meal recalls and records. Fourth, our findings could be applied to a larger population because of the huge sample of participants from various provinces. Last but not least, we had control over a number of possible confounders.

There are additional limitations to this study. First, the calculation of long-term dietary selenium intake using a 3-day, 24-h food survey could be insufficient. Second, the data on sleep duration and other self-reported characteristics may be subject to recall bias among respondents. Third, even though we made adjustments for a variety of sociodemographic and the variables of style of living in the multivariate analysis, including income, personal BMI, and diet, there may still be some other confounding variables. Fourth, there was no information on Se indicators (such as hair and serum Se), sleep quality, or other micronutrients that can affect the ideal length of sleep. Lastly, it is also important to consider other factors that may impact sleep time.

## 5. Conclusions

In conclusion, a high-selenium diet may promote optimal sleep time, and a strong positive association between Se intake and the optimal sleep duration was only found amongst overweight participants. The role of BMI in the relationship between Se intake and optimal sleep duration requires further investigations.

## Figures and Tables

**Figure 1 nutrients-15-00397-f001:**
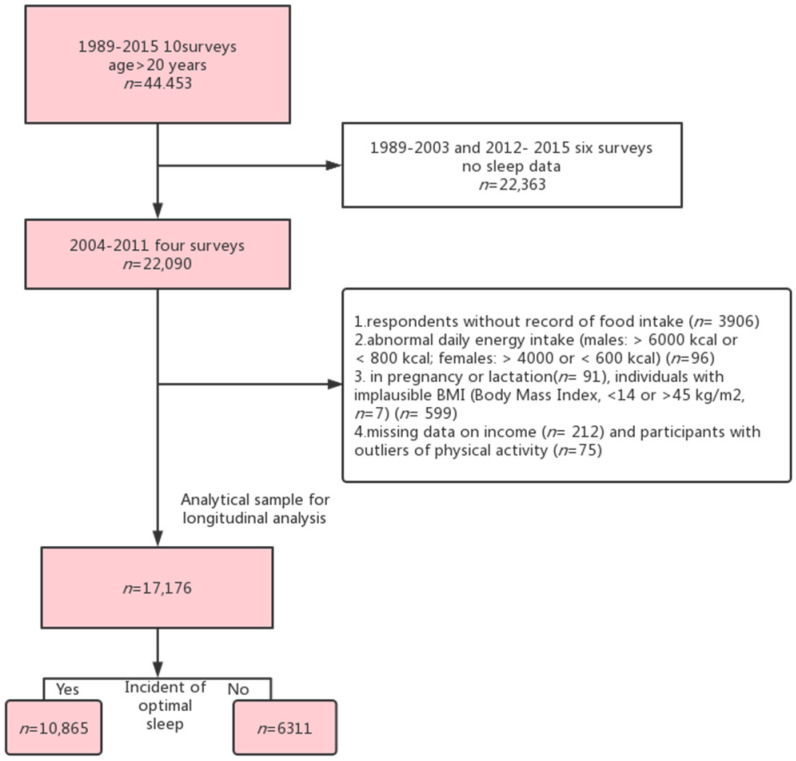
Flow chart of participants in the Selenium Intake and Sleep Survey.

**Figure 2 nutrients-15-00397-f002:**
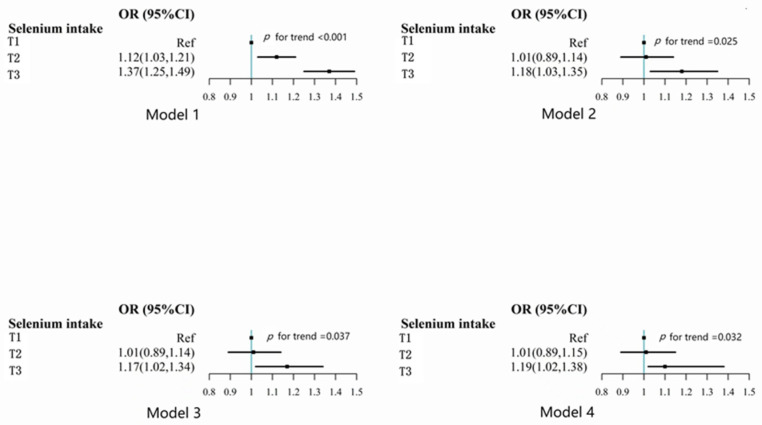
Odds ratios (95% CI) for optimal sleep duration by selenium intake tertiles among Chinese individuals aged 20 or older from China Health and Nutrition Survey. Age, gender, and calorie intake were modified in Model 1. Smoking and alcohol use, income (low, medium, and high), urbanization (low, medium, and high), education (low, medium, and high), and level of physical activity (continuous) were all further accounted for in Model 2. BMI and hypertension were still further modified in Model 3. Self-reported diabetes, fat intake, and energy intake were further modified in Model 4.

**Table 1 nutrients-15-00397-t001:** Baseline sample characteristics of Chinese adults attending CHNS by tertiles of cumulative selenium intake (*n* = 17,176).

Factors	T1	T2	T3	*p*-Value
	*n* = 6380	*n* = 5009	*n* = 5787	
Se intake (µg/day), mean (SD)	23.9 (6.1)	39.2 (3.8)	68.0 (44.8)	<0.001
Energy intake (kcal/day), mean (SD)	1716.3 (536.3)	2072.4 (566.5)	2464.7 (669.6)	<0.001
Fat intake (g/day), mean (SD)	55.3 (30.8)	71.0 (32.5)	87.3 (40.4)	<0.001
Protein intake (g/day), mean (SD)	48.9 (14.5)	65.3 (16.2)	86.3 (26.2)	<0.001
Carbohydrate intake (g/day), mean (SD)	252.6 (100.8)	288.4 (107.3)	325.9 (118.5)	<0.001
Age (years), mean (SD)	48.5 (16.3)	46.4 (15.0)	45.3 (14.4)	<0.001
BMI (kg/m^2^), mean (SD)	22.9 (3.5)	23.3 (3.4)	23.8 (3.4)	<0.001
Gender				<0.001
Men	2482 (38.9%)	2363 (47.2%)	3359 (58.0%)	
Women	3898 (61.1%)	2646 (52.8%)	2428 (42.0%)	
Smoking				<0.001
Non-smoker	4632 (72.8%)	3434 (68.7%)	3602 (62.3%)	
Ex-smoker	232 (3.6%)	168 (3.4%)	253 (4.4%)	
Current smoker	1503 (23.6%)	1393 (27.9%)	1929 (33.4%)	
Income				<0.001
Low	3039 (47.6%)	1720 (34.3%)	1683 (29.1%)	
Medium	1892 (29.7%)	1696 (33.9%)	1916 (33.1%)	
High	1449 (22.7%)	1593 (31.8%)	2188 (37.8%)	
Urbanization				<0.001
Low	2701 (42.3%)	1473 (29.4%)	1497 (25.9%)	
Medium	1797 (28.2%)	1560 (31.1%)	1703 (29.4%)	
High	1882 (29.5%)	1976 (39.4%)	2587 (44.7%)	
SURVEY YEAR				<0.001
2004	3355 (52.6%)	2483 (49.6%)	2750 (47.5%)	
2006	725 (11.4%)	628 (12.5%)	756 (13.1%)	
2009	706 (11.1%)	645 (12.9%)	850 (14.7%)	
2011	1594 (25.0%)	1253 (25.0%)	1431 (24.7%)	
Education				<0.001
Low	4876 (76.6%)	3324 (66.5%)	3450 (59.7%)	
Medium	698 (11.0%)	749 (15.0%)	1032 (17.9%)	
High	792 (12.4%)	923 (18.5%)	1297 (22.4%)	
Hypertension	1379 (21.6%)	987 (19.7%)	1091 (18.9%)	<0.001
Diabetes	156 (2.5%)	132 (2.6%)	149 (2.6%)	0.90
Alcohol drinking	1727 (27.1%)	1671 (33.4%)	2452 (42.4%)	<0.001
Region				<0.001
South	3991 (62.6%)	3040 (60.7%)	3080 (53.2%)	
North	2389 (37.4%)	1969 (39.3%)	2707 (46.8%)	
Overweight	2151 (33.7%)	1963 (39.2%)	2550 (44.1%)	<0.001
Physical activity (MET-h/week), mean (SD)	168.3 (96.6)	156.4 (91.1)	153.5 (86.7)	<0.001
Optimal sleep duration	3860 (60.5%)	3251 (64.9%)	3754 (64.9%)	<0.001

Data are reported as mean (SD) for continuous variables and *n* (%) for categorical variables.

**Table 2 nutrients-15-00397-t002:** Subgroup analyses of selenium consumption with the optimal sleep in China Nutrition and Health Study (*n* = 17,176).

	T1	T2		T3		*p* Trend	*p* for Interaction
Income							0.563
Low	1.00	0.99	(0.81–1.21)	1.19	(0.95–1.48)	0.129	
Medium	1.00	1.08	(0.89–1.31)	1.29	(1.05–1.59)	0.015	
High	1.00	1.01	(0.73–1.40)	1.16	(0.83–1.63)	0.287	
Urbanization							0.168
Low	1.00	1.00	(0.85–1.18)	1.18	(0.98–1.42)	0.097	
Medium	1.00	1.27	(1.02–1.57)	1.31	(1.04–1.65)	0.023	
High	1.00	0.74	(0.51–1.07)	1.04	(0.71–1.53)	0.385	
Education							0.549
Low	1.00	1.04	(0.90–1.19)	1.19	(1.02–1.38)	0.021	
Medium	1.00	1.37	(0.95–1.99)	1.39	(0.95–2.02)	0.107	
High	1.00	0.70	(0.44–1.10)	1.27	(0.78–2.06)	0.125	
Region							0.226
South	1.00	1.07	(0.92–1.26)	1.37	(1.15–1.64)	<0.001	
North	1.00	1.11	(0.90–1.36)	1.15	(0.94–1.42)	0.226	
Gender							0.766
Men	1.00	1.01	(0.83–1.24)	1.36	(1.10–1.67)	0.001	
Women	1.00	1.13	(0.96–1.34)	1.22	(1.02–1.47)	0.017	
Overweight							0.018
No	1.00	1.07	(0.92–1.24)	1.21	(1.03–1.43)	0.128	
Yes	1.00	1.14	(0.91–1.42)	1.51	(1.19–1.91)	<0.001	
Hypertension							0.647
No	1.00	1.04	(0.90–1.20)	1.23	(1.06–1.43)	0.005	
Yes	1.00	1.26	(0.98–1.61)	1.64	(1.25–2.15)	<0.001	

Odds ratios (with a 95% confidence interval) are the values from multilevel mixed-effects logistic regression. Age, gender, caloric intake, smoking, alcohol use, income, urbanization, education, physical activity, BMI, hypertension, and self-reported diabetes were all taken into account in multilevel mixed-effects logistic regression models. In the appropriate models, stratification factors were not altered. Based on the tertiles of annual income, the distribution of income was divided as low, middle, and high.

## Data Availability

The datasets created and analyzed for the current work may be found in the CHNS repository at https://www.cpc.unc.edu/projects/china (accessed on 18 April 2022).

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
