# Peer review of "Association between Selenium Intake and Optimal Sleep Duration: A National Longitudinal Study"

_nutrients, 2023, doi:10.3390/nu15020397_

Round 1
Reviewer 1 Report
This paper describes the correlation between Se intake and self-report sleep across multiple years in a robust population. The study design and analysis are appropriate. Some English editing would help the reader understand the paper better. The only other suggestion I have is to depict the dose-response associated between Se intake and sleep duration as a graph rather than as a table. I think seeing the relationship graphically would be more impactful.
Author Response
This paper describes the correlation between Se intake and self-report sleep across multiple years in a robust population. The study design and analysis are appropriate. Some English editing would help the reader understand the paper better. The only other suggestion I have is to depict the dose-response associated between Se intake and sleep duration as a graph rather than as a table. I think seeing the relationship graphically would be more impactful.
Response 1: Corrected.
Reviewer 2 Report
Below are my comments
Introduction
1. In the introduction you somehow made the jump from sleep extension and sleep deprivation to using selinium. You need a better transition
2. Is there any data on how selinium would impact sleep? If there is theorized mechanistic effects of selinium please add those to the introduction as well. Just talking about the role of selinium as an anti-oxidant does not quite cut it.
Methods
1. Although I tried to find the sleep questionnaire I was unable to. Can you please identify which survey had the sleep questionnaire?
2. Did you calculate Se intake or was that already calculated witht he data that you had access to?
Otherwise great job with explaining your methodology
Results
1. The results are well written
Discussion
1. Very well written discussion
Author Response
Introduction:
Point 1: Lines 31-32:
The authors would mention this point. In the introduction you somehow made the jump from sleep extension and sleep deprivation to using selinium. You need a better transition.
Response 1: Inserted.
Point 2:
Is there any data on how selinium would impact sleep? If there is theorized mechanistic effects of selinium please add those to the introduction as well. Just talking about the role of selinium as an anti-oxidant does not quite cut it.
Response 2: Thanks for your comments. There is a lack of data and theoretical mechanisms on how selenium affects sleep, so I could only explain the effect of selenium on sleep through its antioxidant effects.
Methods
Point 1:
Although I tried to find the sleep questionnaire I was unable to. Can you please identify which survey had the sleep questionnaire?
Response 3: Thanks for your comments. You can find sleep questionnaire in personal survey of CHNS, the number is U324.
Point 4:
Did you calculate Se intake or was that already calculated witht he data that you had access to? Otherwise great job with explaining your methodology.
Response 4: We calculated the selenium intake based on the food intake and the Chinese Food Composition Table [1].
Reference:
- Hu, F.B.; Stampfer, M.J.; Rimm, E.; Ascherio, A.; Rosner, B.A.; Spiegelman, D.; Willett, W.C.
Reviewer 3 Report
The authors examined the relationship between dietary selenium (Se) intake and sleep duration among 17,176 Chinese adults enrolled in the China Health and Nutrition Survey (CHNS) from 2004 to 2011. As a result, the Se intake was significantly associated with many characteristics (such as energy intake, BMI, smoke, or alcohol habit and overweight). The statistical analysis using fully adjusted regression models revealed that no relationship was found among the four groups classified by quartiles of Se intake. Of interest, a strong positive association between Se intake and the optimal sleep duration was found only among overweight participants.
The findings obtained by this study are considerably of interest and the manuscript is well-written. However, several points need clarifying. These are given below.
Major comments:
1. This is the most critical points. The authors used the cumulative (average) Se intake as parameters for analysis, however, which is not appropriate. Because this study used the data of CHNS from 2004 to 2011, sleep duration and Se intake would be varied depending on survey year. If this treatment was done to “reduce variation amongst individuals (as described by the authors)”, the sleep duration should be also averaged. And the evidence that “cumulative Se intake represented long-term eating habits” should be provided by the Se intake in each survey year. If the Se intake of each participant at 2004, 2006 or 2009 was same (not significant), using the averaged data is reasonable. While, if it was different, the relationship between Se intake and sleep duration should be analyzed separately in survey year. The authors would mention this point.
2. Because I am not familiar with the multilevel mixed-effect logistic-regression analysis, I would like the authors to explain the validity of this model. For the adjustment of potential confounding factors, the four models were built in the Table 2 and the subgroup analysis was done under adjustment of age, gender, intake of energy etc. in Table 3. But I wonder whether or not this model building algorithm was validated and strictly selected. For example, stepwise algorithm is generally adopted in the multivariate analysis. Thus, I would recommend that the detail and validity of model building algorithm would be opened.
3. In the relation with Comments #1 and #2, the nutritional parameters including energy intake, BMI and overweight were strongly associated with Se intake (Table 1). Thus, the fact that the high Se consumer can get optimal sleep is possibly an epiphenomenon of high energy intake (e.g. postprandial somnolence due to blood glucose spike). And the high intake of micronutrient other than Se may contribute to optimal sleep duration. Please refer it.
4. In general, overweight and obesity have led to poor quality of sleep. But the present study indicated that overweight people with high Se intake get optimal sleep duration. I guess that this finding means that poor quality sleep only last long in high BMI participants. Thus, the authors would mention not only the quantity of sleep but also the quality of it.
5. In this study, the authors adopted the definition of overweight as BMI ≥ 24 kg/m2. However, this definition is different from that of WHO (BMI ≥ 25). The authors should explain this point.
6. I don’t know what the words, “5,200 optimal sleep cases at baseline were obtained” means (in page 2, line 88; in Figure 1). For all results, the number of participants was 17,176 in Table 1, 2, and 3. Where is the data of 5,200 sleep cases used?
Minor comments:
1) in page 2, lines 83 to 87, the words, “were excluded” should be added.
2) in page 4, lines 155 to 161, the significant relationship between age and Se intake could be mentioned.
3) in Table 2, the rows, “Global cognitive function”, “Model 5”, and “Model 6” were uncertain.
4) in page 7, line 184, “BMI” was corrected to “overweight”.
5) in page 7, the legend of Table 3 should be rewritten in order to clearly understand this Table indicated the subgroup analysis under multilevel mixed-effects logistic regression models.
6) in the whole text, the words, “optimal(double-spaced)sleep duration” were existed. Please correct it.
I hope these comments will be helpful.
Round 2
Reviewer 2 Report
I appreciate the authors addressing my concerns.
Author Response
Thank you very much for your help!
Reviewer 3 Report
The manuscript has been revised well. I think this manuscript will be acceptable.
Author Response
Thank you very much for your help!